# Artificial Intelligence-Based Methods and Omics for Mental Illness Diagnosis: A Review

**DOI:** 10.3390/bioengineering12101039

**Published:** 2025-09-27

**Authors:** Glenda Santos de Oliveira, Fábio Henrique dos Santos Rodrigues, João Guilherme de Moraes Pontes, Ljubica Tasic

**Affiliations:** Laboratory of Biological Chemistry, Institute of Chemistry, Universidade Estadual de Campinas, Campinas 13083-970, SP, Brazil; glendaso@unicamp.br (G.S.d.O.); rodriguesfhs@gmail.com (F.H.d.S.R.); ljubica@unicamp.br (L.T.)

**Keywords:** personality disorders, psychiatric nosology, biomarker discovery, multi-omics, artificial intelligence, learning architectures, explainability

## Abstract

The underlying causes fof major mental illnesses, including anxiety disorders (ADs), depression, and bipolar disorder (BD), remain insufficiently understood, limiting the availability of effective, patient-friendly treatments and accurate diagnostic tests. For instance, anxiety disorders encompass a diverse spectrum of subtypes and may emerge at different stages of mental illness, each with distinct symptom profiles. This heterogeneity often complicates differential diagnosis, leading, in many cases, to delayed treatment or inappropriate management. In recent years, technological advances have enabled the development of artificial intelligence (AI)-based approaches that, when integrated with multi-omics data, offer substantial advantages over traditional statistical methods, particularly for analysing large-scale datasets and integrating clinical with bioanalytical information. This review analyses current efforts to identify biomarkers for mental illness and explores the application of machine learning, deep learning, and computational modelling in advancing personalised and precise diagnostics.

## 1. Introduction

Personality disorder is characterised by a persistent, inflexible pattern of inner experience and behaviour that deviates from cultural expectations, causing constant distress or impairments [1,2]. The most recent revision of the Diagnostic and Statistical Manual of Mental Disorders, Fifth Edition (DSM-5) [3] includes disturbances in identity, self-direction, empathy, and intimacy, based on psychodynamic theory [4,5].

Traditional diagnostic methods for identifying personality disorders rely on clinical assessments, including structured and semi-structured interviews [6]. While diagnostic frameworks and assessment tools have advanced over the past decades, current approaches remain limited in their capacity to provide early and accurate identification. This is particularly problematic given that personality disorders often present complex symptom patterns and high rates of comorbidity. As a result, many cases remain undetected or are incorrectly diagnosed until symptoms become severe and entrenched, delaying effective intervention and increasing the risk of chronic impairment [7].

Addressing these limitations requires the development of novel and integrative diagnostic strategies that extend beyond subjective evaluation. In recent years, a variety of emerging technologies, such as neuroimaging, multimodal data fusion, and digital phenotyping, have shown potential for capturing real-time, objective markers of personality pathology. At the same time, omics sciences, particularly genomics, proteomics, and metabolomics, have enabled systematic characterisation of biological systems at multiple molecular levels, offering insight into the biological mechanisms of mental illness [8].

The concept of omics is related to systematic methodologies aimed at comprehensively understanding a given biological system, such as cells, tissues, or organs, by measuring variations in genes (genomics), proteins (proteomics), and metabolites (metabolomics) [9]. The integration of artificial intelligence (AI) with multi-omics data offers a transformative opportunity for psychiatric diagnosis [10]. The complexity and heterogeneity of omics datasets, however, demand analytical tools capable of integrating diverse data types and identifying subtle, nonlinear patterns that traditional statistical methods may overlook. Artificial intelligence (AI), particularly deep learning, offers this capability. By combining clinical, molecular, and biological information, AI–omics approaches can reveal biomarker signatures and diagnostic indicators that would otherwise remain hidden [11]. This integration holds the promise of improving diagnostic precision, facilitating earlier detection, and enabling more personalised mental health care.

In this review, we examine current advances in applying AI to multi-omics data for the diagnosis of mental illness, with particular attention to their potential in overcoming the diagnostic challenges of personality disorders.

## 2. Methods

A comprehensive literature search was conducted between 2015 and 2025 using the databases Google Scholar, Scopus, PubMed, and Web of Science. The search strategy combined controlled vocabulary and free-text terms related to biomarkers, mental illness, and artificial intelligence. The primary search terms included the following: biomarkers; mental diseases; biomarkers of depression; biomarkers of bipolar disorder; biomarkers of anxiety disorders; artificial intelligence for mental disease diagnostics; AI in psychiatry; multi-omics biomarkers in mental illness; machine learning in psychiatric diagnosis; explainable AI mental health; psychiatric biomarker discovery; computational psychiatry. Boolean operators were applied to combine concepts, such as “biomarkers” AND “bipolar disorder” AND “artificial intelligence”, and filters were set to include peer-reviewed original research articles, review articles, and meta-analyses published in English.

The inclusion criteria were

Studies involving human participants or clinically relevant human-derived samples (blood, urine, CSF, neuroimaging data, etc.).Research explicitly addressing biomarker discovery, validation, or application for diagnosis or prognosis of depression, bipolar disorder, or anxiety disorders.Studies applying artificial intelligence, machine learning, deep learning, or computational modelling in the analysis of psychiatric biomarkers, particularly within an omics framework.

The exclusion criteria were

Studies involving only animal models without translational validation.Articles not directly addressing diagnosis or biomarker discovery in psychiatric disorders.Conference abstracts, editorials, and opinion pieces without original data.

The titles and abstracts of all retrieved articles were screened by at least two independent reviewers, with disagreements resolved by consensus. Full-text articles meeting the inclusion criteria were analysed and relevant data extracted, including disorder studied, sample type, biomarker(s) identified, omics platform used, AI methodology applied, and main findings.

## 3. Artificial Intelligence, Machine Learning, and Deep Learning

In clinical diagnostics, artificial intelligence (AI) has been applied to interpret large datasets related to health states [12]. This is often achieved through AI frameworks, originally developed in computational engineering, which provides a foundation for building and extending specific functionalities [13]. These frameworks support the design of AI algorithms, commonly used to process persistent datasets and statistical data [12].

In the last decades, the development of AI has occurred in parallel with metabolomics. Since the 1970s, AI-based techniques have been used on chemical data [13]. However, it was from the 1990s onwards that AI began to be employed in machine learning (ML), comprising Principal Component Analysis (PCA), Bayesian methods, Support Vector Machines (SVMs), Random Decision Forest, Decision Trees, k-nearest neighbours (k-NN), and Linear Discriminant Analysis (LDA). Posteriorly, sophisticated deep learning (DL), such as Deep Neural Networks (DNN), Deep Belief Networks (DBN), Recurrent Neural Networks (RNN), Deep Boltzmann Machines (DBMs), Convolutional Neural Networks (CNNs), Autoencoders, and Generative Adversarial Networks (GANs) [14,15], have been used.

Chemometrics, ML, and DL are AI-based methods that have changed the practice of chemistry, particularly in data processing and analysis [13,16]. Chemometrics differs from learning-based methods as it relies on linear relationships within datasets, whereas ML involves training algorithms capable of handling large, complex, and non-linear datasets [17]. DL, a subset of ML, uses neural networks (NN) for data extraction [18].

Compared with ML, DL offers advantages, such as performing classification and prediction directly on raw data (end-to-end learning), reducing potential bias, enabling greater data abstraction, and integrating diverse data types, including text, numerical values, and images [15]. These architectures are essential for handling big data, given the rapid accumulation of omics datasets [19].

Beyond statistical analysis and model training, AI can also enhance different stages of omics workflows and facilitate multi-omics data integration [20]. In metabolomics, AI can be incorporated into different steps of metabolite detection, data extraction, and preprocessing [14]. In proteomics, it has been used as a tool for faster processing of large datasets, more reliable clinical predictions, and improved accuracy in protein identification [21]. In genomics, AI has also been applied to extract large datasets and identify relevant features within genetic sequences. Moreover, it has also supported the development of software designed to automate predictions in large-scale DNA and RNA specificities, predict chromatin effects, and reconstruct three-dimensional genome structures [22]. Figure 1 summarises AI contributions to omics and clinical diagnosis across the following areas: data abstraction, prediction, multi-omics integration, and computational modelling.

Figure 1 shows that there is an implicit connection between AI and omics, since AI-based methods allow a greater integration of the big dataset generated by different techniques, speeding up the process of biomarker annotation, performing the pattern recognition in images, and make a very important part of the development of computational models, which are used in omics.

## 4. Computational Modelling

Computational modelling has been applied in various fields, including the simulation of biological systems for testing experimental conditions [23], metabolic engineering [24], metabolic flux analysis [25], and genome-scale metabolic models (GEMs), which facilitate multi-omics data integration [26]. Furthermore, it has also been explored as a potential diagnostic tool for diseases, such as aortic aneurysms [27], ophthalmological diseases [28], and heart valve disease [29].

In psychiatry, which is a growing field with limited studies, computational modelling has been used to investigate the relationship between neural activity and behaviour [30,31]. This is closely linked to reinforcement learning (RL) models, in which individuals seek rewards by learning through trial and error in challenging environments. Such data can inform the design of predictive algorithms [30,32]. One approach is the prediction error (PE), calculated by comparing expected and actual outcomes, yielding positive or negative PE values. In this sense, Hauser et al. (2019) demonstrated how social PE data from adults could be applied in computational psychiatry, as these data help update self-esteem variables potentially linked to depression and anxiety [30,33].

Despite its potential, computational modelling in psychiatry faces challenges, including the complexity of mathematical formulations, difficulties in interpretation, and the tendency of RL models to attribute variance to a limited set of parameters, which may lead to generalisations. Nonetheless, it offers a valuable framework to connect psychological constructs, neural activity, and behaviour [31].

## 5. Making the Invisible Visible: The Role of Explainable AI in Mental Health Diagnostics

### 5.1. What Is Explainability and Why It Matters

Explainable AI (XAI) is a set of methods that make machine learning outputs interpretable, offering insight into how models reach their predictions. Unlike traditional “black-box” models, where only inputs and outputs are visible, XAI reveals internal decision processes [34,35]. In high-stakes fields such as healthcare, interpretability is essential not only for ethical reasons but also to build trust among patients and professionals. Clinicians, psychologists, and counsellors are more likely to adopt AI as a diagnostic assistant when they understand how a model arrives at its conclusions, enhancing and supporting human decision-making [36,37].

In psychiatry, where diagnoses often involve overlapping symptoms and subjective assessment, the need for explainable models is even greater. XAI enables auditability, helping detect algorithmic bias or discrimination, and ensuring fairness toward underrepresented groups [38]. Legal compliance is also a driver; for instance, the EU General Data Protection Regulation (GDPR) grants individuals a “right to explanation,” requiring deployed models to provide interpretable outputs [39].

### 5.2. Explainability in Practice: What Techniques Are Used

To implement XAI, a range of techniques can be used (Figure 2). These methods reveal different aspects of a machine learning model, such as which features influence predictions most or the reasoning behind a decision. Key techniques include the following:

SHAP (SHapley Additive exPlanations) uses cooperative theory to provide both global and local interpretability and works across model types, from random forests to deep neural networks. It is, however, computationally intensive and harder to interpret when many features have weak contributions [40].

LIME (Local Interpretable Model-agnostic Explanations) offers fast, intuitive, local explanations at lower computational cost but is less stable, more sensitive to perturbations, and less effective when features are highly correlated. It also provides patient-specific explanations for individual classifications, enhancing the potential for complex classifiers to act as safety aids in clinical settings [41].

Decision Trees—These supervised algorithms can serve as surrogate models to approximate complex systems in an interpretable way, clearly visualising decision pathways. They are a reliable and effective decision-making technique, providing high classification accuracy and a simple representation of gathered knowledge. Decision trees have been applied across various areas of medical decision-making and can aid clinicians or policymakers by offering transparent logic. Conceptually simple decision-making models with the capacity for automatic learning are often the most appropriate for such tasks, though decision trees may still oversimplify complex, non-linear relationships [42].

Saliency Maps—These highlight which regions of an image or spectrum most influence predictions, providing intuitive visual overlays. While useful, they can be noisy, occasionally misleading, and lack a ranking of feature importance [43].

Grad-CAM—Gradient-weighted Class Activation Mapping extends saliency maps for convolutional neural networks (CNNs), generating class-specific heatmaps to show which regions drive predictions. More robust than standard saliency maps, it can highlight, for example, anatomical regions relevant to a diagnosis. Its use is, however, limited to CNN architecture and is less suitable for non-spatial data such as tabular inputs [44].

## 6. Multi-Omics and AI in Psychiatry

Psychiatry and omics data are notably challenging to analyse. Some of their characteristics prove to be difficult to address, such as their high dimensionality. Omics samples can have hundreds or thousands of features (multiple metabolites, peaks in spectra, genes, etc.), resulting in very complex architectures, specially in deep neural networks. Other important aspects are the high collinearity and sparsity of psychiatric and omics data, which makes it challenging to isolate signals, and little variation across subjects, thus making feature attribution more challenging [45].

The complex nature of this type of data often requires deep learning (DL) architectures to be able to handle these complexities more efficiently than traditional ML methods [46]. On the other hand, it results in very complex models, with thousands (or more) parameters, which make XAI challenging to perform [47]. This results in a paradox, in which omics systems require more complex DL models that can detect subtle or weak signals, but in turn, these diminish the model’s interpretability.

Psychiatric diagnoses are much more nuanced and probabilistic than other types of diagnoses and often are not binary. In this context, clinicians need to be able to provide evidence that justify their diagnostic decisions, so they can offer clarity for patients and their families, to be accountable to regulatory bodies and ethical review boards and medical insurance providers [48]. Interpretability in ML can build trust, which is crucial in psychiatry, fostering shared decision-making and tailored interventions [49]. The goal of XAI in psychiatry is to augment and support human judgement, not replace, acting as a bridge to detect subtle patterns that may elude clinicians while still grounding decisions in human expertise [48].

Although XAI is a promising approach for psychiatry and multi-omics, it still requires further development to address its main limitations. Future research will focus on ML models that are both accurate and interpretable by design, incorporating biological elements to reduce opacity in high-dimensional data. Developing personalised XAI tailored to individual patient profiles, rather than generalised models, could advance precision psychiatry and customised treatments.

## 7. From Multi-Omics to Machine Learning: Advances in Psychiatric Diagnostics

Misdiagnosis or delayed diagnosis of mental illnesses often leads to inadequate treatment and can exacerbate the patient’s condition. Despite the incomplete understanding of their pathogenesis, current diagnostic practices still rely largely on clinical assessment, psychiatric evaluation, and standardised rating scales [50,51]. Psychiatric disorders are the result of multifactorial causes; therefore, different multi-omics studies are done in an attempt to obtain a greater understanding of the biochemical mechanisms involved. So, AI-based methods as DL are employed for the integration and organisation of these big datasets.

The integration of artificial intelligence (AI) with multi-omics data offers powerful tools for detecting complex, non-linear patterns that traditional methods often miss [52]. This is particularly relevant for conditions such as depression, bipolar disorder, and anxiety disorders [53,54], which rank among the most prevalent and disabling mental illnesses worldwide. These disorders share overlapping symptoms, display substantial clinical heterogeneity, and frequently co-occur, making accurate diagnosis especially challenging [55,56]. In these contexts, applying AI to multi-omics datasets can enable earlier detection, improve differential diagnosis, and ultimately guide more personalised treatment strategies, as illustrated in Figure 3.

In this review, we focus on three major categories of mental health disorders: bipolar disorder, anxiety disorders, and depression.

### 7.1. Bipolar Disorder

Bipolar disorder (BD) is a chronic and recurrent mood disorder characterised by shifts in mood, thinking, behaviour, energy, and the ability to carry out daily activities [57]. According to the Global Burden of Disease report (2019), the lifetime prevalence of bipolar disorder is estimated to be between 0.4% and 1.1% worldwide [58]. Additionally, the World Health Organization (WHO) reports that the disorder affects approximately 45 million people [59].

The diagnosis of bipolar disorders (BDs) remains challenging, as depressive episodes can closely resemble those of unipolar depression [60]. This overlap, along with complex and variable symptomatology, contributes to frequent misdiagnosis and delays in treatment. However, the wide range of possible symptom combinations results in substantial clinical heterogeneity [61,62].

BDs have a multifactorial origin involving biological, genetic, and environmental influences. Factors such as serotonin and norepinephrine imbalance, elevated cortisol levels, hypothalamic–pituitary–adrenal (HPA) axis dysfunction, neurohormonal irregularities, and genetic predisposition are all linked to their onset and progression. While the exact cause remains unclear, these biological alterations are considered central to BD pathophysiology [63].

The absence of definitive biomarkers means diagnosis still relies heavily on clinical evaluation and long-term follow-up [64]. In recent years, studies have explored artificial intelligence-driven analyses of genetic, proteomic, and metabolomic data to identify potential biomarkers for BDs, aiming to support earlier and more accurate diagnosis.

Genome-wide association studies (GWASs) analyse hundreds of thousands to millions of genetic variants across the genomes of many individuals to identify those statistically associated with a specific trait or disease [65]. Recent GWAS investigations have identified over 60 loci linked to bipolar disorder (BD), particularly genes involved in synaptic signalling and highly expressed in prefrontal cortex and hippocampal neurons, including HTR6, MCHR1, DCLK3, and FURIN [66]. BD-associated genes participate in diverse biological processes, such as cellular signalling (notably cAMP and calcium pathways: CACNA1C, CAMK2A, CAMK2D, ADCY1, ADCY2) and neurotransmission via glutamatergic (GRIK1, GRM3, GRM7), dopaminergic (DRD2, DRD4, COMT, MAOA), and serotonergic (HTR1A, HTR2A, HTR3B) systems. Genes involved in neuronal growth and development, including BDNF, IGFBP1, NRG1, and NRG3, are also strongly implicated in BD pathophysiology [67]. Moreover, chromosomes 5 [68], 11 [69], and X [70] show strong associations with the disorder.

Increasing evidence highlights the role of multiple biological dysfunctions in BD, particularly systemic inflammation, with elevated levels of pro-inflammatory cytokines, such as TNF-α, IL-6, sTNF-R1, and sIL-2R, observed in patients [71,72]. Similarly, mitochondrial dysfunction and oxidative stress have been implicated, as impaired mitochondrial function leads to excessive reactive oxygen species (ROS) production, causing damage to lipids, proteins, and DNA [73]. These converging pathways underline the multifactorial nature of BD and suggest potential avenues for biomarker development and targeted interventions.

Complementarily, by identifying changes in metabolites and analysing disrupted metabolic pathways, metabolomics offers potential for improving diagnostic accuracy and deepening our understanding of the disorder’s biological basis [74]. More specifically, metabolomic studies have identified a range of altered metabolites in serum, plasma, and cerebrospinal fluid, highlighting disruptions in amino acid, lipid, carbohydrate, and energy metabolism. In serum, upregulation of 3-hydroxyacetic acid, *N*-acetyl-glycoprotein, mannose, alanine, and dimethylglycine, along with downregulation of glucose, lactate, acetoacetate, acetate, and ascorbate, has been reported across different BD states (mania, hypomania, depression, and mixed episodes) [75,76]. Similarly, plasma and serum analyses indicate increased kynurenine and tryptophan metabolites, as well as alterations in phospholipids and amino acids [77,78]. Meanwhile, cerebrospinal fluid studies further confirm dysregulation of kynurenine, kynurenic acid, tryptophan, and picolinic acid [79]. Collectively, these findings point to consistent metabolic perturbations in BD that may serve as potential biomarkers and provide insight into underlying pathophysiological mechanisms. Table 1 summarises some biomarkers of some mental illnesses found in omics studies reported in the literature.

Delving deeper into the use of AI in diagnostic support for BD, recent advances in artificial intelligence have highlighted the potential of large language models (LLMs) to aid clinical decision-making in psychiatry. For example, Perlis et al. (2024) evaluated LLMs as decision-support tools for the pharmacological treatment of bipolar depression [80]. The study compared a standard LLM with one augmented using evidence-based guidelines. While LLMs can rapidly synthesise large volumes of information, they still lack access to the most up-to-date research and the nuanced clinical judgment of human practitioners. However, augmenting the model with guideline-based data significantly improved the validity and reliability of its recommendations.

Although artificial intelligence (AI) has already been applied to enhance BD management, improving mood episode prediction, personalising treatment plans, and providing real-time support [81], it has not yet been fully leveraged for multi-omics studies. Integrating AI with genomic, proteomic, and metabolomic data could unlock additional insights into BD pathophysiology.

Campos-Ugaz et al. (2023) demonstrated machine learning applications in diagnosing bipolar disorder using EEG and brain imaging data [82]. Furthermore, Sundaram et al. (2017) developed a deep convolutional neural network model for predicting bipolar disorder from genomic data, motivated by the disorder’s heritability. The architecture automatically extracted genotype features to classify bipolar phenotypes and achieved top performance in the bipolar disorder challenge [83]. Interestingly, Shen (2024) developed a bioinformatics–machine learning diagnostic model to distinguish bipolar disorder (BD) from schizophrenia (SC) and major depressive disorder (MDD) using brain tissue [84]. Among other things, RBM10 emerged as a key marker differentiating BD from SC, while LYPD1, HMBS, HEBP2, SETD3, and ECM2 distinguished BD from MDD. Immune cell infiltration analysis revealed significant differences in B cells, NK cells, and mast cells between BD and the other disorders, supporting the potential of these biomarkers for improving BD diagnostic precision.

### 7.2. Anxiety Disorders

Anxiety disorders are marked by feelings of fear, uncertainty, apprehension, often triggered by perceived threats and more intense symptoms, including unexpected panic attacks and social avoidance [85,86]. Anxiety disorders are categorised as generalised anxiety disorder (GAD), agoraphobia (AP), social anxiety disorder (SAD) or social phobia, panic disorder (PD), and separation anxiety disorder [87,88]. The onset of different anxiety disorders occurs at distinct developmental stages. For example, specific phobias tend to emerge around the age of 7, social anxiety disorder (SAD) around 13, agoraphobia (AP) around 20, panic disorder (PD) around 24, and generalised anxiety disorder (GAD) typically manifests after the age of 50 [89].

Anxiety disorders (ADs), ranked by the World Health Organization as the 6th leading cause of global disability [90], affecting an estimated 301.4 million people worldwide between 1990 and 2019, a prevalence higher than that of schizophrenia and bipolar disorder [91]. In Europe, mood disorders are estimated to cause an average annual economic loss of EUR 113.4 billion [92]. Globally, depression and anxiety disorders result in productivity losses of about USD 1 trillion per year, with projections indicating this could rise to USD 16 trillion by 2030 [93].

Recent studies have identified key genes associated with anxiety disorders through blood-based epigenetic analyses. Ciuculete et al. (2018) highlighted the STK32B gene to differentiate between low-risk and high-risk individuals for generalised anxiety disorder (GAD) [94]. In a broader approach, Kwon et al. (2024) analysed 17 methylation biomarkers, including MUTYH, TOE1, MIR3146, DIP2C, SEC23IP, INPP5A, ESRRA, and ZNF689, among others, in AD patients [95]. Additionally, Zou et al. (2023) focused on the CBL gene to distinguish panic disorder (PD) cases. These findings underscore the utility of epigenetic biomarkers in blood as promising tools for understanding the molecular mechanisms underlying anxiety disorders [96].

Similar to bipolar disorder, significant progress has been made in developing AI-based applications to enhance the management, diagnosis, and treatment precision of anxiety disorders. For example, Oliveira et al. (2025) developed an AI-based Self-Care app to enhance mental health self-awareness and facilitate communication with clinicians, tracking symptom and stress triggers, generating evidence-based reports on anxiety patterns [97]. In another study, Manole et al. (2024) assessed an AI-powered chatbot, built with ChatGPT, for managing mild to moderate anxiety using evidence-based cognitive-behavioural therapy techniques, offering constantly available real-time support, demonstrating potential for accessible anxiety symptom management [98].

Building on the promising use of AI, particularly large language models, in supporting psychiatric diagnosis and treatment decisions, there is growing interest in applying AI to analyse complex biological data from omics studies. Understanding the molecular and genetic underpinnings of AD through omics approaches can provide new insights into their pathophysiology.

Liu (2021) applied neural network models to whole-genome sequencing cohorts of individuals with Down syndrome with confirmed anxiety disorders, revealing 17q25, 16q23, 21q22, and 22q13, and 29 as recurrent anxiety-specific biomarkers and offering new opportunities for targeted diagnosis and intervention in vulnerable groups [99]. Complementing this genomic focus, a recent study detected generalised anxiety disorder using audio and questionnaire data. A hybrid ResNet-like convolution neural network (CNN) architecture, achieving 81.23% accuracy, demonstrated the potential of advanced deep learning methods for timely, non-invasive anxiety assessment [100]. Together, these studies illustrate how cutting-edge AI approaches can uncover population-specific molecular signatures and enhance precision in diagnosing and managing anxiety disorders.

### 7.3. Depression

Depressive disorders are among the most prevalent and disabling mental health conditions, affecting approximately 300 million people worldwide [101,102]. The main consequences of depression include memory and cognitive function loss, aging, increased risk of obesity, frailty, and diabetes and may lead the individual to commit suicide [103,104,105]. Reports by Fu et al. (2021) indicated that major depressive disorder had the highest suicide rate among individuals with mental disorders (bipolar disorder, major depressive disorder, and schizophrenia) at 534.3 per 100,000 person-years. Suicide accounts for 1.4% of all deaths worldwide [103], underscoring the importance of identifying biomarkers of depression early in life.

Omics studies attempt to find biomarkers of depression, for example, Jiang et al. (2022) detected bilirverdin as a male-specific biomarker and phosphatidylcholine (10:0/14:1) as a female-specific biomarker in children and adolescents with major depressive disorder (MDD) [106]. Furthermore, Pan et al. (2018) assessed plasma neurotransmitters in individuals with MDD and bipolar disorder (BD) reporting alterations in metabolites, such as succinic acid, γ-aminobutyric acid (GABA), glutamine, α-ketoglutaric acid, L-tyrosine, dopamine, and kynurenine [107]. From a proteomics perspective, Bot et al. (2015) identified alterations in several proteins and hormones, including pancreatic polypeptide, follicle-stimulating hormone, apolipoprotein D, and insulin growth factor-binding protein [108].

Integrating ML to detect major depression, Bouzid et al. (2023) revealed that MDD-associated genes are enriched in immune, inflammatory, neurodegeneration, and cerebellar atrophy pathways [109]. The most relevant genes included NRG1, CEACAM8, CLEC12B, DEFA4, HP, LCN2, OLFM4, SERPING1, TCN1, and THBS1. Among these, NRG1 (linked to synaptic plasticity and neurotransmission) emerged as the most robust discriminator across diverse external datasets and was validated in saliva samples from an independent cohort. Functional brain mapping demonstrated high NRG1 expression in subcortical limbic regions implicated in depression, supporting its potential as a non-invasive biomarker for early MDD detection.

Beyond that, Squarcina et al. (2021) applied deep learning to predict treatment response in depression using patient datasets [110]. Deep learning demonstrated strong potential, often outperforming traditional regression methods and achieving accuracies of around 80%, particularly when integrating neuroimaging with clinical and molecular biomarkers. However, small sample sizes and limited interpretability remain major challenges. Expanding dataset sizes and enhancing model transparency are essential steps toward enabling deep learning-driven personalised treatment strategies in depression.

At last, the potential biomarkers of mental health diseases, such as, anxiety disorders, depression, and bipolar disorder, identified in human samples in different omics studies from 2015—2025, are summarized in Table 1.

**Table 1 bioengineering-12-01039-t001:** Potential biomarkers of anxiety disorders, depression, and bipolar disorder identified in human samples in different omics studies.

Disorder	Potential Biomarkers	Sample Type	Measurement	Sampling *	Reference
	Metabolomics/Lipidomics				
Anxiety	*N*-methyl nicotinamide, amino malonic acid, azelaic acid, and hippuric acid	Urine	NMR and GC-MS	AD with or without depression (*N* = 48) and HC (*N* = 48)	Chen et al., 2018 [111]
Anxiety	Glycoprotein acetyls, docosahexaenoic acid, serum total triglycerides, omega-3 fatty acids, apolipoprotein B, VLDL cholesterol, and glucose, among others (13 metabolites)	Plasma	NMR	AD + depression (*N* = 531), depression (*N* = 304), AD (*N* = 548), remitted depression and/or AD (*N* = 897), and HC (*N* = 634)	Kluiver et al., 2021 [112]
Anxiety	Increased histidine, 2-phenylacetamide, cytosine, and 4-hydroxy hippuric acid; decreased glutamine, threonic acid, L-methionine, malic acid, L-valine, citric acid, tyrosine, and propionyl carnitine, among others (22 metabolites)	Serum	LC-MS	AD (*N* = 18) and HC (*N* = 31)	Vismara et al., 2020 [51]
Anxiety	Uridine, 3-methoxy tyrosine, L-methionine, L-leucine, and xanthine, among others (43 metabolites)	Serum	UPLC-MS/MS	PD (*N* = 55) and HC (*N* = 55)	Shan et al., 2023 [113]
Depression	57 male-related and 53 female-related differential metabolites, where biliverdin was a male-specific biomarker and phosphatidylcholine (10:0/14:1) was a female-specific biomarker	Plasma	UPLC-Q-TOF/MS	MDD (*N* = 84—42 male and 42 female) and HC (*N* = 49—27 male and 22 female)	Jiang et al., 2022 [106]
Depression and Bipolar	MDD: succinic acid, *γ*-aminobutyric acid, glutamine, *α*-ketoglutaric acid, L-tyrosine, tyramine, dopamine, tryptophan, and kynurenine	Plasma neurotransmitters	LC-MS/MS and GC-MS	MDD (*N* = 49), BD (*N* = 30), and HC (*N* = 40)	Pan et al., 2018 [107]
Bipolar	Nervonic acid, 4-phenylbutyric acid, and 1-methyluric acid	Plasma	UPLC-MS/MS-/Orbitrap	BD (*N* = 91 and HC (*N* = 92)	Wei et al., 2021 [114]
Bipolar	Betaine, glycerol, hippuric acid, indole sulphate, trimethylamine oxide (upregulated), and inositol (downregulated)	Urine	NMR	BD (*N* = 37) and HC (*N* = 48)	Ren et al., 2021 [115]
Bipolar and Depression	DD: 2-hydroxyhippuric acid, tyramine-O-sulfate, and isobutyryl-L-carnitine (upregulated); 4-hydroxyphenylacetylglycine, vanilloylglycine, and L-cysteinylglycine disulfide (downregulated) BD: 2-aminoisobutyric acid, dopamine, pyridoxal, hexanoylglycine, citric acid, 3-hydroxysebacic acid, leucylproline, *N*-undecanoylglycine, norepinephrine sulfate, and hexanoylcarnitine (Upregulated) Shared: urobilin, hypoxanthine, guanine, serotonin, tyrosine, 4-pyridoxic acid, *N*-acetyl-L-glutamic acid, and dihydroxyindole, (upregulated)	Urine	UHPLC-MS	DD (*N* = 50), BD (*N* = 12), and HC (*N* = 50)	Wang et al., 2024 [116]
Bipolar	BH, BE, and BM: 3-hydroxyacetic acid, *N*-acetyl-glycoprotein, and mannose (upregulated) BH and BE: glucose (upregulated); lactate and acetoacetate (downregulated) BM: alanine and dimethylglycine (upregulated); acetate and ascorbate (downregulated)	Serum	NMR	BD and depressive episodes (BE, *N* = 59), BD and mania/hypomania episodes (BH, *N* = 16), BD and mixed episodes BM (*N* = 10), and HC (*N* = 10)	Guo et al., 2024 [76]
Bipolar	Threonine, aspartate, gamma-aminobutyric acid, 2-hydroxybutyric acid, serine, mannose, 3-hydroxybutyric acid, arginine, lysine, tyrosine, phenylalanine, glycerol, lactate, alanine, valine, leucine, isoleucine, glutamine, glutamate, glucose, and choline	Serum	NMR	BD (*N* = 33) and HC (*N* = 39)	Simić et al., 2023 [75]
Bipolar	Kynurenic acid, kynurenine, tryptophan, and picolinic acid (upregulated)	CSF	UPLC-MS/MS	BD (*N* = 101) and HC (*N* = 80)	Trepci et al., 2021 [79]
	Proteomics				
Anxiety	Receptor tyrosine kinase (AXL), vascular cell adhesion molecule-1 (VCAM-1, a membrane protein), and 3 other proteins of patients with comorbidities	Serum	Multi-analyte profiling immunoassay platform	SAD or depression (*N* = 2329) and HC (*N* = 652)	Gottschalk et al., 2015 [117]
Anxiety	Brain-derived neurotrophic Factor (BDNF) protein	Serum	Enzyme-linked immunosorbent assay kits	PD (*N* = 90) and HC (*N* = 99)	Li, J. et al., 2023 [118]
Anxiety	Proteins related to the immune response and GAD: A1BG, C4-A, TF, V3-3, and DEFA1	Serum	Tandem mass tags (TMT) combined with HPLC-MS/MS	drug-naïve GAD (*N* = 21) and HC (*N* = 22)	Li, X et al., 2024 [119]
Depression	Pancreatic polypeptide, follicle-stimulating hormone (FSH), prostatin, angiogenin, apolipoprotein D, luteinising hormone, *α*-1-antitrypsin, *α*-1-antichymotrypsin, macrophage migration inhibitory factor, growth-regulated *α*-protein, insulin growth factor-binding protein-5, etc.	Serum	Multiplexed microbead immunoassays	current MDD (*N* = 687), remitted MDD (*N* = 482), and HC (*N* = 420)	Bot et al., 2015 [108]
Bipolar	Platelet-Derived Growth factor BB (PDGF-BB) and thrombospondin-1 (TSP-1)	Blood	Multiplex immunoassay	BD (*N* = 70), MDD (*N* = 42), and HC (*N* = 18)	Kittel-Schneider et al., 2020 [120]
Bipolar	BD and MDD: C3 (upregulated). BD and MDD: C4BPα and CFI (downregulated)	Plasma	Two-dimensional electrophoresis (2-DE and MALDI-TOF/TOF MS	BD (*N* = 20), MDD (*N* = 30), and HC (*N* = 30)	Chen et al., 2015 [121]
Bipolar	BD: serotransferrin, vanin, apolipoprotein A-I; endoglin, suprabasin; sulfhydryl oxidase (upregulated); alpha-1-acid glycoprotein, C-type mannose receptor 2, antileukoproteinase (downregulated)	Serum	LC-MS/MS	BD (*N* = 30) and HC (*N* = 30)	Ren et al., 2017 [122]
Bipolar	CLEC1B, TNFRSF21, tenascin-R, disintegrin and metalloproteinase domain-containing protein 23, cell adhesion molecule 3, RGM domain family member B, plexin-B1, brorin, CACNG4, and PLIN5	CSF	LC-MS/MS	Two independent case-control cohorts (total *N* = 351)	Göteson et al., 2021 [123]
Bipolar	IL6, MCP-1, TGF-α, IL8, β-NGF, and IL10-RB (upregulated)	Plasma	ELISA/Olink proteomics	BD (*N* = 32) and HC (*N* = 16)	Xu et al., 2024 [124]
	Genomics/Transcriptomics				
Anxiety	*STK32B* gene	Blood	Microarray technique (methylation)	Low-risk of GAD (*N* = 164) and high-risk of GAD (*N* = 55)	Ciuculete et al., 2018 [94]
Anxiety	Genes *MUTYH*, *TOE1*, *MIR3146*, *DIP2C*, *SEC23IP*, *INPP5A*, *ESRRA*, and *ZNF689*, among others (17 methylation biomarkers)	Blood	Bootstrapping techniques	AD (*N* = 94) and HC (*N* = 296)	Kwon et al., 2024 [95]
Depression	*ABCA13* rs4917029, *BNIP3* rs9419139, *CACNA1E* rs704329, *EXOC4* rs6978272, *GRIN2B* rs7954376, *LHFPL3* rs4352778, *NELL1* rs2139423, *NUAK1* rs2956406, *PREX1* rs4810894, and *SLIT3* rs139863958	Analysis of genetic and clinical factors	Deep learning technique—multilayer feedforward neural networks (MFNNs).	MDD (*N* = 455)	Lin et al., 2018 [125]
Bipolar	FADS1/2 gene	Brain	LC-MS/MS	*Fads1*/*2* knockout mice	Yamamoto et al., 2023 [78]
Bipolar	Found 64 associated genomic loci, including CACNB2, KCNB1, BTN2A1, HTR6, MCHR1, DCLK3, and FURIN	Brain	GWAS	BD (*N* = 41,917) and HC (*N* = 371,549)	Mullins et al., 2021 [66]
Bipolar	ANK3, CACNA1C, CACNA1B, HOMER1, KCNB1, MCHR1, NCAN, and SHANK2	Brain	GWAS	-	Li et al., 2021 [126]
Bipolar	TRANK1	Brain	GWAS and eQTL	BD (*N* = 1784) and HC (*N* = 2474)	Li et al., 2021 [127]
Bipolar	298 genome significant loci, including FURIN, MED24, THRA, ALDH2, ANKK1, ARHGAP15, CACNA1B, ERBB4, ESR1, FES, GPR139, HTT, MLEC, MSH6, PSMD14, TOMM2 ALDH2, ESR1, HTT, ERBB4, CACNA1B, SHANK2, OLFM1, SHISA9, SORCS3, and LR5NF	Brain	GWAS	BD (*N* = 158,036) and HC (*N* = 2.8 million)	O’Connell et al., 2025 [128]
Bipolar and Depression	SYT14	Brain	GWAS	BD (*N* = 1822) and HC (*N* = 4650); MDD (*N* = 5303) and HC (*N* = 5337);	Zhang et al., 2025 [129]
Bipolar	CRMP1, SYT4, UCHL1, ADCY1, and ZBTB18	Brain	Single-cell RNA sequencing (scRNA-seq) and GWAS	RNA-seq (*N* = 8, 1266 cells and B GWAS data (*N* = 413,466)	Wei et al., 2024 [130]

* AD—anxiety disorder; GAD—general anxiety disorder; HC—healthy control; MDD—major depressive disorder; PD—panic disorder; SAD—social anxiety disorder; BD—bipolar disorder; GWAS—genome-wide association study; eQTL—expression quantitative trait loci; CSF—cerebrospinal fluid; ELISA.

## 8. Perspectives, Challenges, and Limitations

The growing application of AI-based methods in life sciences and clinical research has motivated the development of advanced tools capable of transforming biomedical decision-making, diagnostics, and drug development [131,132]. Key advantages include the following: (1) increased use of imaging and neuroimaging for biomarker identification [132,133]; (2) ranking and prioritisation of biomarker predictors via machine learning [88]; and (3) the potential replacement of animal models with AI technologies [134]. Nevertheless, limitations remain, including overfitting of data, bias generation, and the need for complementary research to validate AI-selected biomarkers [135].

Machine learning-based biomarker selection also enables the classification of groups and subgroups of anxiety disorders and other neurological illnesses using large-scale datasets [88,136,137]. However, building robust models requires prior knowledge of relevant variables and high-quality datasets to ensure reliable training and trustworthy predictions. Compared to animal models, which involve ethical concerns, high costs, and limited translational value due to anatomical and physiological differences, AI-based in silico approaches may better predict biochemical responses to treatments [138].

Despite these advantages, significant challenges remain for the clinical application of AI. The complexity of AI methods demands multidisciplinary expertise and a thorough understanding of the mental disorder under study [132,134]. Many deep learning models operate as “black boxes”, offering limited interpretability of their predictions, since not all variables used in training are specified. Explainable AI approaches are emerging to provide human-understandable interpretations, addressing the opacity of traditional models [138]. Additionally, biological systems are dynamic, and algorithms require continuous retraining to accommodate new data and maintain accuracy [134].

The possibility of application of AI-based methods in studies of psychiatric disorders is still a challenge in routine clinical practice due to the fact that there is no anatomical difference in the patient’s brains. However, there are examples of searches using magnetic resonance imaging (MRI) for analysis of glioma samples and employing artificial intelligence (chatGPT-4V) for the recognition of profiles in diagnosis [139]. Another example has been the use of machine learning in the ranking of breast cancer drugs and biomarker identification [140]. These searches have increasingly motivated the employment of AI tools in the neurology area.

In psychiatry, AI implementation is particularly challenging because patients often lack clear differences, complicating diagnosis. The presented advances indicate that, while full integration of AI into clinical practice remains a challenge, ongoing methodological improvements and expanded datasets are steadily moving the field forward.

## Figures and Tables

**Figure 1 bioengineering-12-01039-f001:**
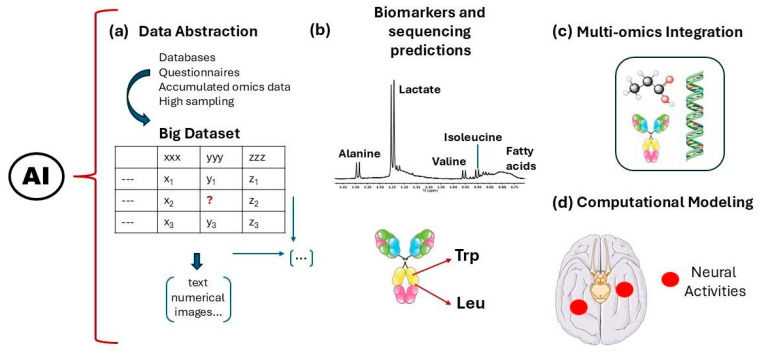
Advantages of AI applications in omics data: (**a**) abstraction of large datasets for statistical analysis and prediction, including estimation of missing values; (**b**) acceleration of biomarker annotation and prediction of DNA, RNA, and protein sequences through recognition of pre-established patterns; (**c**) multi-omics integration, enabling the correlation of data from different omics layers; (**d**) computational modelling for the simulation of biological systems. Images of DNA, immunoglobulin, and brain were adapted from smart.servier.com (free medical images).

**Figure 2 bioengineering-12-01039-f002:**
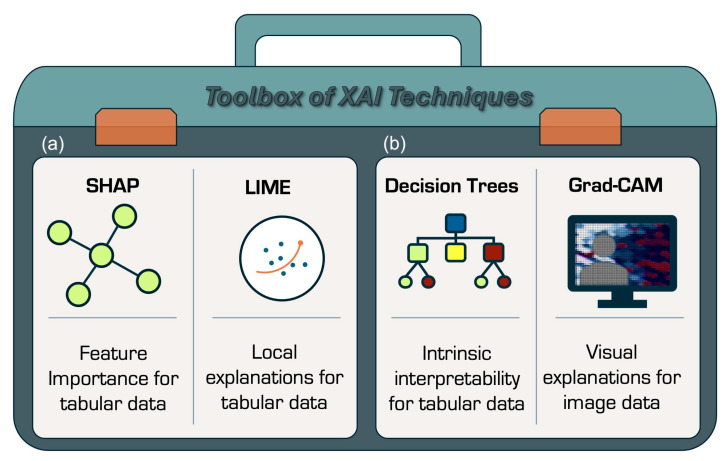
Illustration of toolbox of Explainable Artificial Intelligence (XAI) techniques, (**a**) SHapley Additive exPlanations (SHAP) and Local Interpretable Model-agnostic Explanations (LIME), which are feature-attribution techniques; (**b**) Decision Trees used in surrogate models as supervised learning algorithms and Gradient-weighted Class Activation Mapping (Grad-CAM) used in analysis of imaging.

**Figure 3 bioengineering-12-01039-f003:**
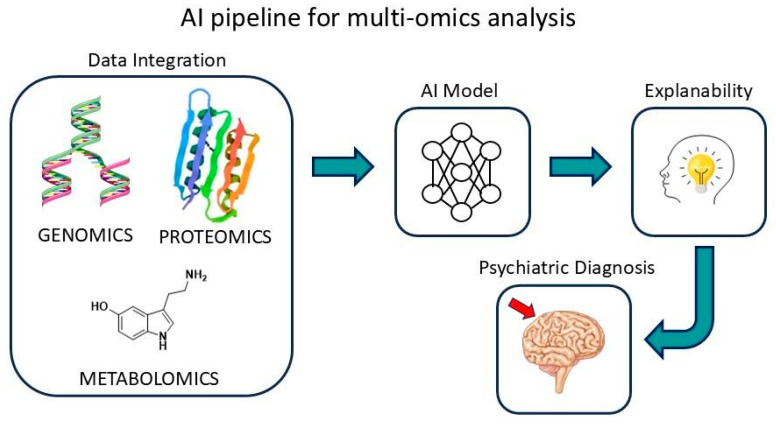
AI pipeline for multi-omics analysis. Large-scale data acquired through different techniques are integrated into an AI model, processed by algorithms, and transformed into an interpretable output. This information can then be applied to the development of diagnostic methods.

## Data Availability

No new data were created or analyzed in this study. Data sharing is not applicable to this article.

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
