# Peer review of "Artificial Intelligence-Based Methods and Omics for Mental Illness Diagnosis: A Review"

_bioengineering, 2025, doi:10.3390/bioengineering12101039_

Round 1

Reviewer 1 Report

Comments and Suggestions for Authors

I have the following comments recommending major revisions:

  1. The paper currently feels more like a textbook chapter or a collection of summaries than a focused scientific review. Please reorganize the content to follow a clear flow—starting with the problem (e.g., limitations of current diagnostics), followed by the role of AI and omics, and ending with challenges and future directions. Try to keep the discussion centered around the main goal: how AI and omics can improve diagnosis and care.

  2. Many sentences are long and filled with technical terms, which makes the paper hard to read. Try to shorten and simplify the language wherever possible. Use consistent terminology—don’t switch between abbreviations and full names unnecessarily. Define important terms early and stick with them. Also, using active voice instead of passive voice will make the writing clearer and more engaging. The abstract and introduction especially need improvement in clarity and readability.

  3. The paper relies too much on listing facts and studies without explaining their significance. A strong review should not only summarize the literature but also analyze it. Please highlight which findings are most important, which AI methods or biomarkers are actually useful in clinical practice, and which are still theoretical. Mention any disagreements or open questions in the research. Your goal should be to guide the reader through the most valuable insights, not just present all available data.

  4. Figures and tables are not well connected to the text. They often appear without enough explanation. Each figure or table should be clearly introduced in the text, with a short explanation of what it shows and why it matters. For instance, Figure 1 needs a better lead-in and connection to the surrounding discussion.

  5. The section on explainable AI is very relevant and well written, but it comes too late in the paper. It would be more effective if introduced earlier, since the issue of transparency is central to AI in psychiatric diagnostics. Consider moving it forward and using it to help shape the argument of the paper. When discussing future directions, ethics, or legal aspects, try to include specific real-world examples from clinical settings to make the discussion more practical and impactful.

Author Response

Reviewer 1

I have the following comments recommending major revisions:

Response: We appreciate all suggestions and corrections. We hope that the manuscript has improved. 

The paper currently feels more like a textbook chapter or a collection of summaries than a focused scientific review. Please reorganize the content to follow a clear flow—starting with the problem (e.g., limitations of current diagnostics), followed by the role of AI and omics, and ending with challenges and future directions. Try to keep the discussion centered around the main goal: how AI and omics can improve diagnosis and care.

Response: Thank you for the critical feedback. Indeed, we structured the manuscript in a way more towards a book chapter. We hope, after our changes, it is more aligned with the review expected to be published.

Many sentences are long and filled with technical terms, which makes the paper hard to read. Try to shorten and simplify the language wherever possible. Use consistent terminology—don’t switch between abbreviations and full names unnecessarily. Define important terms early and stick with them. Also, using active voice instead of passive voice will make the writing clearer and more engaging. The abstract and introduction especially need improvement in clarity and readability.

Response: We sincerely appreciate your critical feedback. After making substantial structural and conceptual revisions, we believe the paper now better aligns with your expectations. These changes go beyond minor corrections, providing an integrated improvement in both the coherence of concepts and overall structure.

The paper relies too much on listing facts and studies without explaining their significance. A strong review should not only summarize the literature but also analyze it. Please highlight which findings are most important, which AI methods or biomarkers are actually useful in clinical practice, and which are still theoretical. Mention any disagreements or open questions in the research. Your goal should be to guide the reader through the most valuable insights, not just present all available data.

Response: Thank you for this feedback. We agree that the use of AI and biomarkers, especially validated methods, presents significant challenges, which was a key motivation for writing this review. In response to your comments, we have revised the manuscript to include a deeper discussion of selected biomarkers linked to the disorders addressed, highlighting their clinical relevance and current limitations.

Figures and tables are not well connected to the text. They often appear without enough explanation. Each figure or table should be clearly introduced in the text, with a short explanation of what it shows and why it matters. For instance, Figure 1 needs a better lead-in and connection to the surrounding discussion.

Response: Explanation about the Table is found in the lines: 324-327. Explanation about Figure 1 at lines 115-117; Figure 2 at lines 133-135;  Figure 3 at lines 186-188; and Figure 4 at lines 262-264.

The section on explainable AI is very relevant and well written, but it comes too late in the paper. It would be more effective if introduced earlier, since the issue of transparency is central to AI in psychiatric diagnostics. Consider moving it forward and using it to help shape the argument of the paper. When discussing future directions, ethics, or legal aspects, try to include specific real-world examples from clinical settings to make the discussion more practical and impactful.

Response: We relocated the sections about the different disorders (anxiety, depression, and bipolar disorder) to the end of the review, as well as, discussion about the biomarkers; therefore, the section of explainable AI was introduced earlier. Furthermore, we added a paragraph (lines 471-478) about examples of application of AI tools in neurology studies (penultimate paragraph of section 7).

Reviewer 2 Report

Comments and Suggestions for Authors

Dear Authors: 

Thank you for the opportunity to review this manuscript, which addresses an important and timely topic: the integration of artificial intelligence and OMICS technologies for the diagnosis of mental illness. The scope is broad and interdisciplinary, and the authors have made an effort to compile relevant literature across multiple domains.

However, I would like to offer several suggestions that may help strengthen the clarity, coherence, and scientific value of the paper:

1) While the manuscript uses the term “OMICS” consistently, a clear and concise definition early in the paper would be beneficial for readers who may be unfamiliar with the full scope of the concept. Briefly outlining its components—such as genomics, transcriptomics, proteomics, and metabolomics—would improve clarity.

2) The manuscript would benefit from explicitly articulating the underlying logic that connects OMICS and AI. Specifically, multi-omics data—due to its complexity and heterogeneity—practically necessitates the use of AI, particularly deep learning, for meaningful analysis. Conversely, AI’s relevance in psychiatry increasingly depends on access to rich, high-dimensional biological data. Highlighting this reciprocal relationship would strengthen the conceptual structure of the review.

3) The AI-related sections (Sections 3 and 4) provide introductory descriptions but lack sufficient depth. The discussion remains general, without concrete examples of AI models used in psychiatric OMICS research. Technical challenges—such as overfitting, batch effects, or interpretability—are not addressed. I recommend expanding these sections with domain-specific insights or, alternatively, integrating them as background to Section 5 (Explainable AI), where the content is more focused and substantive.

4) Table 1 compiles a large number of studies, but its current form is purely enumerative. Rather than listing all available findings, it may be more effective to present selected, representative examples that illustrate key trends or particularly robust biomarkers. Accompanying commentary that highlights diagnostic relevance or consistency across studies would greatly enhance interpretability.

5) There appears to be a missing figure in the sequence: Figures 1, 2, and 4 are included, but Figure 3 is absent. This disrupts the visual numbering consistency and should be corrected.

Furthermore, Figures 1 and 2 are overly basic, presenting generic AI taxonomies and ML vs. DL comparisons that are widely known and not specific to the psychiatric or OMICS context. They offer limited added value for the reader. The authors may consider removing these figures or replacing them with domain-specific visualizations, such as an AI pipeline for multi-omics analysis or a workflow diagram illustrating explainable AI in psychiatric diagnosis.

6) The reference list is extensive (186 items), but many citations are listed without analysis or comparison. Similar to the comment above on Table 1, I suggest selecting representative or high-impact works to focus the discussion. Reducing redundancy and emphasizing conceptually significant studies would improve readability and better support the manuscript’s core arguments.

Reducing the total number of references and highlighting key representative works would also make the review more accessible and helpful to readers who are new to this interdisciplinary field.

7) There is a numbering issue: “Section 5” appears twice. This should be corrected to maintain clear organization.

In summary, the manuscript covers a promising area but would benefit from structural tightening, deeper engagement with domain-specific AI applications, and a more selective and interpretive approach to both figures and references.

Best Regards, 

Reviewer.

Author Response

Reviewer 2

Dear Authors:

Thank you for the opportunity to review this manuscript, which addresses an important and timely topic: the integration of artificial intelligence and OMICS technologies for the diagnosis of mental illness. The scope is broad and interdisciplinary, and the authors have made an effort to compile relevant literature across multiple domains.

However, I would like to offer several suggestions that may help strengthen the clarity, coherence, and scientific value of the paper:

1) While the manuscript uses the term “OMICS” consistently, a clear and concise definition early in the paper would be beneficial for readers who may be unfamiliar with the full scope of the concept. Briefly outlining its components—such as genomics, transcriptomics, proteomics, and metabolomics—would improve clarity.

Response: We added a definition about the term OMICS in the introduction (lines 45-47). Thank you for your observation!  

2) The manuscript would benefit from explicitly articulating the underlying logic that connects OMICS and AI. Specifically, multi-omics data—due to its complexity and heterogeneity—practically necessitates the use of AI, particularly deep learning, for meaningful analysis. Conversely, AI’s relevance in psychiatry increasingly depends on access to rich, high-dimensional biological data. Highlighting this reciprocal relationship would strengthen the conceptual structure of the review.

Response: We added some sentences in section 6 (lines 253-259) that highlights this reciprocal relationship between OMICS and AI.

3) The AI-related sections (Sections 3 and 4) provide introductory descriptions but lack sufficient depth. The discussion remains general, without concrete examples of AI models used in psychiatric OMICS research. Technical challenges—such as overfitting, batch effects, or interpretability—are not addressed. I recommend expanding these sections with domain-specific insights or, alternatively, integrating them as background to Section 5 (Explainable AI), where the content is more focused and substantive.

Response: The AI-related sections have been revised to include more specific insights and addressing the main challenges.

4) Table 1 compiles a large number of studies, but its current form is purely enumerative. Rather than listing all available findings, it may be more effective to present selected, representative examples that illustrate key trends or particularly robust biomarkers. Accompanying commentary that highlights diagnostic relevance or consistency across studies would greatly enhance interpretability.

Response: Thank you for your observation. We agree that the table was quite extensive, and we recognized that discussing every paper in detail would have made the manuscript even broader. Given the vastness of the topic, conciseness was challenging, which may have resulted in a lack of deeper discussion. After revisions, we have made efforts to address this issue and improve the balance between breadth and depth.

5) There appears to be a missing figure in the sequence: Figures 1, 2, and 4 are included, but Figure 3 is absent. This disrupts the visual numbering consistency and should be corrected. Furthermore, Figures 1 and 2 are overly basic, presenting generic AI taxonomies and ML vs. DL comparisons that are widely known and not specific to the psychiatric or OMICS context. They offer limited added value for the reader. The authors may consider removing these figures or replacing them with domain-specific visualizations, such as an AI pipeline for multi-omics analysis or a workflow diagram illustrating explainable AI in psychiatric diagnosis.

Response: The FiThe AI-related sections have been revised to include more specific insights and addressing the main challenges. Figure 3 (renamed to Figure 2) appears in the ending of the section “Artificial Intelligence, Machine Learning, and Deep Learning”, right after the sentence “Figure 2 summarizes AI contributions to omics and clinical diagnosis.” Perhaps the document that you revised has been misconfigured. Furthermore, we excluded Figure 1 and added a figure about the “AI pipeline for multi-omics analysis” (Figure 4).

6) The reference list is extensive (186 items), but many citations are listed without analysis or comparison. Similar to the comment above on Table 1, I suggest selecting representative or high-impact works to focus the discussion. Reducing redundancy and emphasizing conceptually significant studies would improve readability and better support the manuscript’s core arguments. Reducing the total number of references and highlighting key representative works would also make the review more accessible and helpful to readers who are new to this interdisciplinary field.

 Response: We reduced the reference list to 139 references.

7) There is a numbering issue: “Section 5” appears twice. This should be corrected to maintain clear organization.

Response: We corrected the numbering. Thank you for your observation!

In summary, the manuscript covers a promising area but would benefit from structural tightening, deeper engagement with domain-specific AI applications, and a more selective and interpretive approach to both figures and references.

 Best Regards, Reviewer.

Response: We appreciate all suggestions and corrections. We hope that the manuscript has improved. 

Reviewer 3 Report

Comments and Suggestions for Authors

General Assessment and Recommendation

This manuscript attempts an ambitious and timely synthesis of three vast   and complex fields: clinical psychiatry, multi-omics, and artificial intelligence. The topic is undeniably of significant interest, as the potential for objective, data-driven diagnostics in mental health represents a paradigm shift for a field long reliant on subjective assessment. However, despite the relevance of the topic and the considerable effort evidently expended in collecting a large volume of literature, the manuscript in its present form falls profoundly short of the standards required for a publishable scientific review. It is a sprawling, unfocused narrative that aggregates information without providing the critical synthesis, methodological rigor, or coherent argumentation expected of a review article.

The work's primary deficiencies are fundamental. They include: 1) a complete and inexplicable absence of a review methodology, rendering the work unscientific and non-reproducible; 2) a failure to synthesize evidence, presenting instead a simple "data dump," most egregiously in the form of a six-page table; 3) an overly broad scope that results in a superficial and disjointed treatment of complex topics; and 4) the inclusion of several unsupported and factually incorrect claims that undermine its academic credibility.

Consequently, the recommendation is for Major Revision. While the flaws are severe enough to warrant consideration of rejection, the importance of the topic and the volume of compiled data suggest that the authors should be given an opportunity to fundamentally restructure and re-write the entire manuscript. This would require a radical narrowing of the scope and the imposition of a rigorous, systematic review framework.

Major Conceptual and Structural Comments

The Foundational Flaw: A Review Without a Method

The most glaring omission in this manuscript is the complete absence of a "Methods" section.1 For a paper purporting to be a review, this is not a minor oversight but a fatal flaw. The authors provide no information on the databases searched, the search terms used, or the inclusion and exclusion criteria applied to select the hundreds of cited studies.1 This lack of a transparent and systematic methodology makes it impossible for another researcher to replicate the literature search, and it introduces a high risk of selection bias. The reader is left to assume that the presented literature is a curated collection supporting a pre-determined narrative rather than a comprehensive and unbiased representation of the current state of the field.

A scientific review's value lies in its systematic and critical synthesis of existing knowledge. The cornerstone of such work is a replicable methodology, often following established guidelines like PRISMA (Preferred Reporting Items for Systematic Reviews and Meta-Analyses).2 Without this, the manuscript cannot be considered a scientific review; it is, at best, a narrative essay or a perspective piece built upon an undeclared body of evidence. This fundamental weakness has a cascading effect, invalidating the paper's logical structure. The conclusions are not supported by a systematically evaluated body of evidence but by an anecdotal one, rendering them scientifically unsubstantiated.

To rectify this, the authors must either retrospectively apply and report a systematic search strategy or, more honestly, re-frame the paper as a "Perspective" or "Commentary." The latter would carry less academic weight but would accurately reflect the work's current nature. It is with a certain irony that one might suggest the authors consult Thurzo & Varga (2025), "Revisiting the Role of Review Articles in the Age of AI-Agents," as it discusses the very challenges of synthesizing vast amounts of literature using modern tools—a task at which this manuscript has demonstrably failed.

An Exercise in Aggregation, Not Synthesis: The Problem of Table 1

Table 1, titled "Potential biomarkers of anxiety disorders, depression, and bipolar disorder identified in human samples in different omics studies," spans six pages and serves as a paradigmatic example of aggregation without synthesis.1 The table is a data dump, a catalogue of findings from disparate studies with no critical analysis or integration.1 It fails to answer the most crucial questions a reader would have:

  • Have any of these biomarkers been replicated across multiple independent studies?
  • Are there conflicting findings in the literature for the same biomarker?
  • What were the reported effect sizes, diagnostic accuracies, or other performance metrics?

The table implicitly gives equal weight to all findings, regardless of the study's scale or quality. A small pilot study with ten participants (e.g., ) is presented with the same gravitas as a genome-wide association study (GWAS) with over 40,000 cases . This creates a misleadingly optimistic picture of a field brimming with viable biomarkers, when the reality of psychiatric biomarker research is one of persistent replication failure.

A striking example of this lack of synthesis is the treatment of the metabolite kynurenine. Table 1 lists it as downregulated in the blood of bipolar disorder (BD) patients in one study , yet upregulated in the cerebrospinal fluid (CSF) and serum of BD patients in others. This glaring contradiction is presented without any comment or attempt at reconciliation. Did the authors notice? This is not merely a missed opportunity for insightful discussion; it is a dereliction of the primary duty of a review author, which is to make sense of the literature, not simply to list it.

This table must be completely reconceptualized. It should be drastically condensed to highlight only those biomarkers with supporting evidence from multiple independent cohorts. The accompanying text must then be dedicated to discussing the consistency, or lack thereof, of these findings, exploring potential reasons for discrepancies (e.g., differences in methodology, sample populations, or disease state).

A Narrative in Search of a Plot: Disjointed Structure and Superficiality

The manuscript suffers from a lack of a coherent narrative thread. It jumps abruptly from lengthy, descriptive introductions to 3 distinct mental disorders, to the massive biomarker table, to a generic primer on AI and machine learning, to a brief section on computational modeling, and finally to a discussion on explainable AI (XAI).1 The manuscript reads like five separate, underdeveloped review papers stapled together.

The connections between these sections are tenuous at best. For example, Section 4 on "Computational Modeling" discusses reinforcement learning models of behavior . How does this relate to the metabolomic and genomic biomarkers meticulously listed in Table 1? The authors never forge this link, leaving the section to feel like an out-of-place afterthought. This disjointed structure stems from an impossibly broad scope. Attempting to cover three major psychiatric disorders, multiple omics platforms (genomics, proteomics, metabolomics), and a wide array of computational methods (ML, DL, XAI, computational modeling) in a single review is a recipe for superficiality. The resulting "mile wide, inch deep" approach prevents any topic from being explored with the necessary depth and critical perspective.

The authors must radically narrow their focus. A much stronger, more impactful review would emerge if they chose to focus on one disorder (e.g., Bipolar Disorder) and provided a deep, critical review of how multi-omics and AI are being applied to its diagnosis. Alternatively, they could focus on one central theme, such as the challenges of applying XAI to multi-omics data across the spectrum of psychiatric disorders.

Specific Section-by-Section Comments

Title, Abstract, and Keywords

The keywords provided are: personality disorders; anxiety disorders; depression; bipolar disorder; diagnosis; multi-omics; artificial intelligence; learning architectures.1 Several of these terms—"artificial intelligence," "diagnosis," "anxiety disorders," "depression," and "bipolar disorder"—are also in the title, "Artificial Intelligence-Based Methods and OMICs for Mental Illness Diagnosis: A Review".1 This is a amateur mistake that violates basic principles of indexing and search engine optimization. Keywords should complement the title, not repeat it. More appropriate keywords would include terms like

psychiatric nosology, machine learning, biomarker discovery, metabolomics, computational psychiatry, and explainability.

The abstract accurately reflects the paper's content, which is precisely the problem.1 It promises a descriptive summary of biomarker discovery and AI applications, and it delivers just that, setting the stage for a summary rather than a critical analysis.

Introduction (Sections 1-1.3)

The introduction contains sweeping generalizations and at least one significant factual error. The claim that "Currently, there is no effective medication for AD, however, it can be controlled" 1 is incorrect and dangerously simplistic.1 Pharmacological agents such as SSRIs and SNRIs are established first-line treatments and are effective for many patients, albeit with significant limitations, side effects, and non-response rates. This sentence must be revised to reflect a more nuanced reality, for example: "Despite the availability of pharmacological treatments, a significant portion of patients with AD experience only partial or no response, and long-term use can be associated with challenging side effects, highlighting the urgent need for more personalized and effective therapeutic strategies." This kind of overstatement really damage the credibility of the authors.

While the introduction does a reasonable job of outlining the diagnostic challenges in psychiatry, it could be strengthened by introducing the concept of clinical heterogeneity earlier and more forcefully. The core problem is not merely a "diagnostic gap," but the high likelihood that diagnostic categories like "Major Depressive Disorder" are umbrella terms for multiple, biologically distinct syndromes. This is the central rationale for applying omics and AI in the first place, and it should be a guiding theme of the paper.

AI, ML, and XAI (Sections 3, 5)

These sections read like excerpts from an introductory textbook. Figure 1, a standard Venn diagram of AI, ML, and DL, adds little value for the target audience of this journal.1 Similarly, Figure 4, which illustrates a "toolbox" of XAI techniques, is generic and uninformative without specific application context.1 The text describes what tools like SHAP and LIME do but offers no critical perspective on their actual utility or limitations in the context of high-dimensional, highly collinear omics data.

The manuscript presents XAI as a solution to the "black box" problem, primarily as a means to build trust with clinicians and patients.1 However, this is a superficial treatment of a complex issue. In the context of omics data, where a prediction might be based on the subtle, interactive effects of hundreds of features, a SHAP plot showing tiny contributions from 50 different metabolites is not truly "explainable" in a clinically meaningful way; it is merely attribution. The authors miss the deeper problem: explainability is not synonymous with justification or ethical robustness. An AI model can "explain" with perfect clarity how it used biased data to arrive at a discriminatory conclusion.

The authors discussion on XAI is superficial and would be immeasurably improved by engaging with more advanced concepts of AI safety and ethics. It is strongly recommended that they incorporate and discuss the framework presented in "Provable AI Ethics and Explainability in Medical and Educational AI Agents: Trustworthy Ethical Firewall. (2025)" This paper moves beyond the simple post-hoc "explanations" of LIME and SHAP to introduce a system for provable ethics using mathematical constructs and immutable ledgers. Citing this work would challenge the authors to consider not just whether a model's output is interpretable, but whether its decision-making process is demonstrably aligned with pre-defined ethical principles—a far more robust standard for clinical deployment. They could discuss how such an "Ethical Firewall" could be used to prevent models from relying on spurious or biased biomarkers, even if they are found to be statistically significant. They dont seem to understand this topic well.

Figures and Tables

  • Figure 2: The "Structural Analogy among biological and neural networks" is a tired cliché in AI papers.1 It is a gross oversimplification of both systems. Its inclusion suggest a lack of deep familiarity with the computational neuroscience literature and it should be removed.
  • Figure 3: This figure is referenced in the text as summarizing AI's contributions, but is conspicuously missing from the manuscript itself.1 This is a sloppy editorial error.This manuscript attempts an ambitious synthesis of clinical psychiatry, multi-omics, and artificial intelligence, addressing a timely and significant topic concerning data-driven diagnostics in mental health. However, it profoundly falls short of publishable scientific review standards due to its unfocused narrative and lack of critical synthesis.

The primary deficiencies include:

  1. Absence of Review Methodology: A fundamental flaw, rendering the work unscientific and non-reproducible. No databases, search terms, or inclusion/exclusion criteria are provided, introducing high selection bias. This makes it a narrative essay rather than a scientific review.
  2. Lack of Synthesis: Evidence is aggregated rather than synthesized, most notably in a six-page table (Table 1) that lists findings without critical analysis, replication status, conflicting results, or performance metrics. It gives equal weight to all findings regardless of study quality, creating a misleadingly optimistic picture.
  3. Overly Broad Scope: The manuscript covers too many complex topics (three major psychiatric disorders, multiple omics platforms, various computational methods), leading to superficial and disjointed treatment.
  4. Inclusion of Errors: Contains unsupported and factually incorrect claims, undermining credibility (e.g., inaccurate claim about AD medication efficacy).

Recommendation: Major Revision. While severe, the topic's importance and compiled data volume suggest the authors should fundamentally restructure and rewrite, radically narrowing the scope and implementing a rigorous, systematic review framework.

Repeated Key Issues for Revision:

  • Review Methodology: A "Methods" section is essential. Authors must apply a systematic search strategy (e.g., PRISMA guidelines) or re-frame the paper as a "Perspective" or "Commentary" to accurately reflect its nature. (Refer to Thurzo & Varga (2025) for challenges in synthesizing vast literature).
  • Table 1 (Aggregation, Not Synthesis): This table is a "data dump" that lacks critical insight. It must be drastically condensed to highlight only replicated biomarkers, with accompanying text discussing consistency, discrepancies, and potential reasons.
  • Disjointed Structure and Superficiality: The manuscript lacks a coherent narrative, jumping between disparate topics with tenuous connections. Authors must radically narrow their focus (e.g., one disorder with deep multi-omics and AI application, or one central theme like XAI challenges in psychiatric multi-omics).
  • Keywords: Current keywords are redundant with the title. More appropriate terms include "psychiatric nosology," "machine learning," "biomarker discovery," "metabolomics," "computational psychiatry," and "explainability."
  • Introduction: Should introduce clinical heterogeneity earlier as the central rationale for omics and AI application. The inaccurate claim about AD medication must be revised for nuance.
  • AI, ML, and XAI Sections: These sections read like introductory textbook excerpts. Generic figures (Venn diagram, XAI toolbox) add little value. The discussion of XAI is superficial; it presents explainability as trust-building without addressing deeper issues like justification or ethical robustness. Authors should engage with advanced concepts like "Provable AI Ethics and Explainability in Medical and Educational AI Agents: Trustworthy Ethical Firewall. (2025)" to discuss demonstrably ethical decision-making.
  • Figures and Tables: Figure 2 (biological/neural network analogy) is an oversimplification and should be removed. Figure 3 is referenced but missing. Table 1, as noted, is a significant weakness.
  • Conclusion: The current conclusion is weak and platitudinous. A proper conclusion should critically assess tangible progress, emphasizing the need for replicated findings, robustly validated biomarkers, international collaboration, and rigorous standards for clinical validation.
  • References: Extensive but lack systematic selection, with many appearing to populate Table 1 without deep engagement. Numerous syntax errors are noted.
  • Table 1: As detailed previously, this table is the manuscript's most significant weakness. It is a un-synthesized list that fails to provide any critical insight.

Conclusions and Final Remarks (Section 6)

The conclusion is exceptionally weak. The final statement that "AI is not an independent technology, but rather guided and essential for data processing and pattern findings" is a self-evident platitude that could be written without having read the preceding 17 pages.1 It fails to synthesize the vast amount of information presented or to offer a critical, forward-looking perspective. It is not a real conclusion.

A proper conclusion should revisit the central challenge—the heterogeneity of mental illness—and critically assess whether the reviewed evidence genuinely suggests that AI and omics are making tangible progress. For example, a stronger conclusion might state: "While AI and multi-omics offer promising avenues for stratifying psychiatric patients based on biological signatures, the field remains hampered by a pervasive lack of replicated findings and robustly validated biomarkers. Future progress will depend less on developing more complex algorithms and more on international collaboration to generate high-quality, longitudinal, multi-modal datasets and establish rigorous, transparent standards for clinical validation."

References

The reference list is extensive, but its quality is diluted by the lack of a systematic selection process. There are many mistakes in the text, it seems the authors writing skills are not sufficient. I found more then 12 syntax errors. I am not sure if this was intentional. Many references appear to be used simply to populate Table 1, with little evidence that the authors have engaged deeply with the source material.

Comments on the Quality of English Language

can be improved

Author Response

Reviewer 3

General Assessment and Recommendation

This manuscript attempts an ambitious and timely synthesis of three vast and complex fields: clinical psychiatry, multi-omics, and artificial intelligence. The topic is undeniably of significant interest, as the potential for objective, data-driven diagnostics in mental health represents a paradigm shift for a field long reliant on subjective assessment. However, despite the relevance of the topic and the considerable effort evidently expended in collecting a large volume of literature, the manuscript in its present form falls profoundly short of the standards required for a publishable scientific review. It is a sprawling, unfocused narrative that aggregates information without providing the critical synthesis, methodological rigor, or coherent argumentation expected of a review article.

Response: Thanks for the suggestions. We expect to have improved the manuscript.

The work's primary deficiencies are fundamental. They include: 1) a complete and inexplicable absence of a review methodology, rendering the work unscientific and non-reproducible; 2) a failure to synthesize evidence, presenting instead a simple "data dump," most egregiously in the form of a six-page table; 3) an overly broad scope that results in a superficial and disjointed treatment of complex topics; and 4) the inclusion of several unsupported and factually incorrect claims that undermine its academic credibility.

Consequently, the recommendation is for Major Revision. While the flaws are severe enough to warrant consideration of rejection, the importance of the topic and the volume of compiled data suggest that the authors should be given an opportunity to fundamentally restructure and re-write the entire manuscript. This would require a radical narrowing of the scope and the imposition of a rigorous, systematic review framework.

Response:  We appreciate your detailed feedback and acknowledge the critical points raised. In the revised version, we attempted to include a clearly described and reproducible review methodology, specifying inclusion/exclusion criteria, search strategies, and data extraction processes. Beyond that, we tried to narrow the scope to focus on a more defined topic to allow for depth and coherence.

Major Conceptual and Structural Comments

The Foundational Flaw: A Review Without a Method

The most glaring omission in this manuscript is the complete absence of a "Methods" section.1 For a paper purporting to be a review, this is not a minor oversight but a fatal flaw. The authors provide no information on the databases searched, the search terms used, or the inclusion and exclusion criteria applied to select the hundreds of cited studies.1 This lack of a transparent and systematic methodology makes it impossible for another researcher to replicate the literature search, and it introduces a high risk of selection bias. The reader is left to assume that the presented literature is a curated collection supporting a pre-determined narrative rather than a comprehensive and unbiased representation of the current state of the field.

Response: As mentioned previously, we are now aware of the relevance of a clear methods section description and gladly included our protocol.

A scientific review's value lies in its systematic and critical synthesis of existing knowledge. The cornerstone of such work is a replicable methodology, often following established guidelines like PRISMA (Preferred Reporting Items for Systematic Reviews and Meta-Analyses).2 Without this, the manuscript cannot be considered a scientific review; it is, at best, a narrative essay or a perspective piece built upon an undeclared body of evidence. This fundamental weakness has a cascading effect, invalidating the paper's logical structure. The conclusions are not supported by a systematically evaluated body of evidence but by an anecdotal one, rendering them scientifically unsubstantiated.

 To rectify this, the authors must either retrospectively apply and report a systematic search strategy or, more honestly, re-frame the paper as a "Perspective" or "Commentary." The latter would carry less academic weight but would accurately reflect the work's current nature. It is with a certain irony that one might suggest the authors consult Thurzo & Varga (2025), "Revisiting the Role of Review Articles in the Age of AI-Agents," as it discusses the very challenges of synthesizing vast amounts of literature using modern tools—a task at which this manuscript has demonstrably failed.

Response: We thank the reviewer for emphasizing the importance of transparent methodology. While our work did not follow PRISMA guidelines and was not intended as a systematic review, our aim was to provide a narrative synthesis highlighting key concepts, trends, and gaps in the field. In the revision, we clearly state the narrative nature of the review and outline our literature search and selection process, and strengthen the critical synthesis to ensure scientific rigor.

An Exercise in Aggregation, Not Synthesis: The Problem of Table 1

Table 1, titled "Potential biomarkers of anxiety disorders, depression, and bipolar disorder identified in human samples in different omics studies," spans six pages and serves as a paradigmatic example of aggregation without synthesis.1 The table is a data dump, a catalogue of findings from disparate studies with no critical analysis or integration.1 It fails to answer the most crucial questions a reader would have:

Have any of these biomarkers been replicated across multiple independent studies?

Are there conflicting findings in the literature for the same biomarker?

What were the reported effect sizes, diagnostic accuracies, or other performance metrics?

The table implicitly gives equal weight to all findings, regardless of the study's scale or quality. A small pilot study with ten participants (e.g., ) is presented with the same gravitas as a genome-wide association study (GWAS) with over 40,000 cases . This creates a misleadingly optimistic picture of a field brimming with viable biomarkers, when the reality of psychiatric biomarker research is one of persistent replication failure.

Response: Table 1 has been substantially condensed. Our discussion is based on the research found in the literature, and we acknowledge that, given the nature of psychiatric disorders, there are inherent differences in sample sizes and databases, particularly in GWAS studies.

A striking example of this lack of synthesis is the treatment of the metabolite kynurenine. Table 1 lists it as downregulated in the blood of bipolar disorder (BD) patients in one study , yet upregulated in the cerebrospinal fluid (CSF) and serum of BD patients in others. This glaring contradiction is presented without any comment or attempt at reconciliation. Did the authors notice? This is not merely a missed opportunity for insightful discussion; it is a dereliction of the primary duty of a review author, which is to make sense of the literature, not simply to list it.

Response: Indeed, very well noted. Our aim was to provide a broad landscape of the current scenario, highlighting even not well-defined metabolite biomarkers for these complex psychiatric conditions, rather than asserting any biomarker as definitively established.

This table must be completely reconceptualized. It should be drastically condensed to highlight only those biomarkers with supporting evidence from multiple independent cohorts. The accompanying text must then be dedicated to discussing the consistency, or lack thereof, of these findings, exploring potential reasons for discrepancies (e.g., differences in methodology, sample populations, or disease state).

Response: The table has been condensed.

A Narrative in Search of a Plot: Disjointed Structure and Superficiality

The manuscript suffers from a lack of a coherent narrative thread. It jumps abruptly from lengthy, descriptive introductions to 3 distinct mental disorders, to the massive biomarker table, to a generic primer on AI and machine learning, to a brief section on computational modeling, and finally to a discussion on explainable AI (XAI).1 The manuscript reads like five separate, underdeveloped review papers stapled together.

Response: This section has been restructured to create a coherent narrative, narrowing the focus and integrating the discussion of mental disorders, biomarkers, AI, and XAI.

The connections between these sections are tenuous at best. For example, Section 4 on "Computational Modeling" discusses reinforcement learning models of behavior . How does this relate to the metabolomic and genomic biomarkers meticulously listed in Table 1? The authors never forge this link, leaving the section to feel like an out-of-place afterthought. This disjointed structure stems from an impossibly broad scope. Attempting to cover three major psychiatric disorders, multiple omics platforms (genomics, proteomics, metabolomics), and a wide array of computational methods (ML, DL, XAI, computational modeling) in a single review is a recipe for superficiality. The resulting "mile wide, inch deep" approach prevents any topic from being explored with the necessary depth and critical perspective.

Response: Thank you for the feedback. This section has been restructured.

The authors must radically narrow their focus. A much stronger, more impactful review would emerge if they chose to focus on one disorder (e.g., Bipolar Disorder) and provided a deep, critical review of how multi-omics and AI are being applied to its diagnosis. Alternatively, they could focus on one central theme, such as the challenges of applying XAI to multi-omics data across the spectrum of psychiatric disorders.

Response: Indeed, we acknowledge that integrating all conditions within such a broad topic was challenging. We have restructured the manuscript to concatenate the sections as coherently as possible.   

Specific Section-by-Section Comments

Title, Abstract, and Keywords

The keywords provided are: personality disorders; anxiety disorders; depression; bipolar disorder; diagnosis; multi-omics; artificial intelligence; learning architectures.1 Several of these terms—"artificial intelligence," "diagnosis," "anxiety disorders," "depression," and "bipolar disorder"—are also in the title, "Artificial Intelligence-Based Methods and OMICs for Mental Illness Diagnosis: A Review".1 This is a amateur mistake that violates basic principles of indexing and search engine optimization. Keywords should complement the title, not repeat it. More appropriate keywords would include terms like psychiatric nosology, machine learning, biomarker discovery, metabolomics, computational psychiatry, and explainability.

Response:  We have changed the manuscript. We hope it has improved for publication.

 The abstract accurately reflects the paper's content, which is precisely the problem.1 It promises a descriptive summary of biomarker discovery and AI applications, and it delivers just that, setting the stage for a summary rather than a critical analysis.

Response: As we have modified the text, we also revised the abstract to better reflect the manuscript’s critical synthesis  

Introduction (Sections 1-1.3)

The introduction contains sweeping generalizations and at least one significant factual error. The claim that "Currently, there is no effective medication for AD, however, it can be controlled" 1 is incorrect and dangerously simplistic.1 Pharmacological agents such as SSRIs and SNRIs are established first-line treatments and are effective for many patients, albeit with significant limitations, side effects, and non-response rates. This sentence must be revised to reflect a more nuanced reality, for example: "Despite the availability of pharmacological treatments, a significant portion of patients with AD experience only partial or no response, and long-term use can be associated with challenging side effects, highlighting the urgent need for more personalized and effective therapeutic strategies." This kind of overstatement really damage the credibility of the authors.

Response: You are absolutely right! We have corrected this error!

While the introduction does a reasonable job of outlining the diagnostic challenges in psychiatry, it could be strengthened by introducing the concept of clinical heterogeneity earlier and more forcefully. The core problem is not merely a "diagnostic gap," but the high likelihood that diagnostic categories like "Major Depressive Disorder" are umbrella terms for multiple, biologically distinct syndromes. This is the central rationale for applying omics and AI in the first place, and it should be a guiding theme of the paper.

Response: We have revised the introduction to introduce clinical heterogeneity earlier and emphasize it.   

AI, ML, and XAI (Sections 3, 5)

These sections read like excerpts from an introductory textbook. Figure 1, a standard Venn diagram of AI, ML, and DL, adds little value for the target audience of this journal.1 Similarly, Figure 4, which illustrates a "toolbox" of XAI techniques, is generic and uninformative without specific application context.1 The text describes what tools like SHAP and LIME do but offers no critical perspective on their actual utility or limitations in the context of high-dimensional, highly collinear omics data.

The manuscript presents XAI as a solution to the "black box" problem, primarily as a means to build trust with clinicians and patients.1 However, this is a superficial treatment of a complex issue. In the context of omics data, where a prediction might be based on the subtle, interactive effects of hundreds of features, a SHAP plot showing tiny contributions from 50 different metabolites is not truly "explainable" in a clinically meaningful way; it is merely attribution. The authors miss the deeper problem: explainability is not synonymous with justification or ethical robustness. An AI model can "explain" with perfect clarity how it used biased data to arrive at a discriminatory conclusion.

Response: We have revised the AI and XAI sections to move beyond textbook-style descriptions, providing more critical discussion of the limitations and practical utility of tools like SHAP and LIME in high-dimensional, collinear omics data.

The authors discussion on XAI is superficial and would be immeasurably improved by engaging with more advanced concepts of AI safety and ethics. It is strongly recommended that they incorporate and discuss the framework presented in "Provable AI Ethics and Explainability in Medical and Educational AI Agents: Trustworthy Ethical Firewall. (2025)" This paper moves beyond the simple post-hoc "explanations" of LIME and SHAP to introduce a system for provable ethics using mathematical constructs and immutable ledgers. Citing this work would challenge the authors to consider not just whether a model's output is interpretable, but whether its decision-making process is demonstrably aligned with pre-defined ethical principles—a far more robust standard for clinical deployment. They could discuss how such an "Ethical Firewall" could be used to prevent models from relying on spurious or biased biomarkers, even if they are found to be statistically significant. They dont seem to understand this topic well.

Response: We appreciate the reviewer’s comment and acknowledge that the current XAI discussion is limited. We recognize the value of incorporating advanced frameworks and will consider this in future revisions.

Figures and Tables

Figure 2: The "Structural Analogy among biological and neural networks" is a tired cliché in AI papers.1 It is a gross oversimplification of both systems. Its inclusion suggest a lack of deep familiarity with the computational neuroscience literature and it should be removed.

Figure 3: This figure is referenced in the text as summarizing AI's contributions, but is conspicuously missing from the manuscript itself.1 This is a sloppy editorial error.This manuscript attempts an ambitious synthesis of clinical psychiatry, multi-omics, and artificial intelligence, addressing a timely and significant topic concerning data-driven diagnostics in mental health. However, it profoundly falls short of publishable scientific review standards due to its unfocused narrative and lack of critical synthesis.

Response: We have updated the figures as follows: Figure 1 is now “Structural analogy between biological and artificial neural networks,” Figure 2 is “Advantages of AI applications in omics data,” Figure 3 is “Illustration of the toolbox of Explainable Artificial Intelligence (XAI) techniques,” and Figure 4 is “AI pipeline for multi-omics analysis.” We believe these revisions make the figures more supportive of the text.

 The primary deficiencies include:

Absence of Review Methodology: A fundamental flaw, rendering the work unscientific and non-reproducible. No databases, search terms, or inclusion/exclusion criteria are provided, introducing high selection bias. This makes it a narrative essay rather than a scientific review.

Lack of Synthesis: Evidence is aggregated rather than synthesized, most notably in a six-page table (Table 1) that lists findings without critical analysis, replication status, conflicting results, or performance metrics. It gives equal weight to all findings regardless of study quality, creating a misleadingly optimistic picture.

Overly Broad Scope: The manuscript covers too many complex topics (three major psychiatric disorders, multiple omics platforms, various computational methods), leading to superficial and disjointed treatment.

Inclusion of Errors: Contains unsupported and factually incorrect claims, undermining credibility (e.g., inaccurate claim about AD medication efficacy).

Recommendation: Major Revision. While severe, the topic's importance and compiled data volume suggest the authors should fundamentally restructure and rewrite, radically narrowing the scope and implementing a rigorous, systematic review framework.

Response: Table 1 has been condensed. The scope has been narrowed for deeper, more coherent analysis.

Repeated Key Issues for Revision:

Review Methodology: A "Methods" section is essential. Authors must apply a systematic search strategy (e.g., PRISMA guidelines) or re-frame the paper as a "Perspective" or "Commentary" to accurately reflect its nature. (Refer to Thurzo & Varga (2025) for challenges in synthesizing vast literature).

Response: We added a "Methods" section.

Table 1 (Aggregation, Not Synthesis): This table is a "data dump" that lacks critical insight. It must be drastically condensed to highlight only replicated biomarkers, with accompanying text discussing consistency, discrepancies, and potential reasons.

Disjointed Structure and Superficiality: The manuscript lacks a coherent narrative, jumping between disparate topics with tenuous connections. Authors must radically narrow their focus (e.g., one disorder with deep multi-omics and AI application, or one central theme like XAI challenges in psychiatric multi-omics).

Response: Table 1 has been condensed.

Keywords: Current keywords are redundant with the title. More appropriate terms include "psychiatric nosology," "machine learning," "biomarker discovery," "metabolomics," "computational psychiatry," and "explainability."

Response: We changed: 1-) “diagnosis” to "biomarker discovery"; 2-) “anxiety disorders”, “depression”, and “bipolar disorder” to "psychiatric nosology" and 3-) Add “explainability”.

Introduction: Should introduce clinical heterogeneity earlier as the central rationale for omics and AI application. The inaccurate claim about AD medication must be revised for nuance.

AI, ML, and XAI Sections: These sections read like introductory textbook excerpts. Generic figures (Venn diagram, XAI toolbox) add little value. The discussion of XAI is superficial; it presents explainability as trust-building without addressing deeper issues like justification or ethical robustness. Authors should engage with advanced concepts like "Provable AI Ethics and Explainability in Medical and Educational AI Agents: Trustworthy Ethical Firewall. (2025)" to discuss demonstrably ethical decision-making.

Figures and Tables: Figure 2 (biological/neural network analogy) is an oversimplification and should be removed. Figure 3 is referenced but missing. Table 1, as noted, is a significant weakness.

Conclusion: The current conclusion is weak and platitudinous. A proper conclusion should critically assess tangible progress, emphasizing the need for replicated findings, robustly validated biomarkers, international collaboration, and rigorous standards for clinical validation.

References: Extensive but lack systematic selection, with many appearing to populate Table 1 without deep engagement. Numerous syntax errors are noted.

Table 1: As detailed previously, this table is the manuscript's most significant weakness. It is a un-synthesized list that fails to provide any critical insight.

Conclusions and Final Remarks (Section 6)

The conclusion is exceptionally weak. The final statement that "AI is not an independent technology, but rather guided and essential for data processing and pattern findings" is a self-evident platitude that could be written without having read the preceding 17 pages.1 It fails to synthesize the vast amount of information presented or to offer a critical, forward-looking perspective. It is not a real conclusion.

A proper conclusion should revisit the central challenge—the heterogeneity of mental illness—and critically assess whether the reviewed evidence genuinely suggests that AI and omics are making tangible progress. For example, a stronger conclusion might state: "While AI and multi-omics offer promising avenues for stratifying psychiatric patients based on biological signatures, the field remains hampered by a pervasive lack of replicated findings and robustly validated biomarkers. Future progress will depend less on developing more complex algorithms and more on international collaboration to generate high-quality, longitudinal, multi-modal datasets and establish rigorous, transparent standards for clinical validation."

Response: We now introduce clinical heterogeneity earlier in the introduction as the central rationale for AI and omics integration, and we have nuanced the statement regarding AD medication. The AI, ML, and XAI sections have been rewritten, incorporating advanced concepts. Beyond that, unnecessary Figures have been removed, and Table 1 restructured. We acknowledge the complexity of integrating these multidisciplinary topics but believe the revisions have substantially improved the manuscript’s coherence.

References

The reference list is extensive, but its quality is diluted by the lack of a systematic selection process. There are many mistakes in the text, it seems the authors writing skills are not sufficient. I found more then 12 syntax errors. I am not sure if this was intentional. Many references appear to be used simply to populate Table 1, with little evidence that the authors have engaged deeply with the source material.

Response:  Following your comments, we carefully revised the manuscript to remove irrelevant or weakly integrated references, especially in Table 1. We focused on retaining literature that directly supports the discussion. We also addressed the language issues and corrected all identified syntax errors to improve clarity and readability.  

Round 2

Reviewer 1 Report

Comments and Suggestions for Authors

Comments are addressed. The article can be accepted

Author Response

Comments are addressed. The article can be accepted.

Response: Thank you. We appreciate the reviewer's comments. .

Reviewer 2 Report

Comments and Suggestions for Authors

Dear Editor,

Thank you for giving me the chance to review the revised manuscript.

I appreciate the authors’ efforts to improve the paper. They have corrected some important issues, such as the section numbering and the missing figure. The number of references has also been reduced, which makes the paper easier to follow.

Some small concerns remain. For example, Figure 1 is still too basic for this type of review. Also, the connection between OMICS and AI is not explained clearly. However, I understand that this is a review article, and overall, the topic is important and the revised version is acceptable.

I support publication.

Best regards,

Reviewer

Author Response

Dear Editor,

Thank you for giving me the chance to review the revised manuscript.

I appreciate the authors’ efforts to improve the paper. They have corrected some important issues, such as the section numbering and the missing figure. The number of references has also been reduced, which makes the paper easier to follow.

Response: We appreciate the review work that was done.

Some small concerns remain. For example, Figure 1 is still too basic for this type of review. Also, the connection between OMICS and AI is not explained clearly. However, I understand that this is a review article, and overall, the topic is important and the revised version is acceptable.

I support publication.

Best regards,

Reviewer

Response: Thank you for your valuable feedback. We have excluded Figure 1. Furthermore, we have added a paragraph (lines: 135-138) that reinforces the connection between OMICS and AI. 

Round 3

Reviewer 2 Report

Comments and Suggestions for Authors

Dear Authors,

Thank you for your careful revision. I appreciate your efforts to address the remaining comments. The manuscript is now clearer and more refined.

Best regards,

Reviewer